# Intestinal Barrier Dysfunction and Microbial Translocation in Patients with First-Diagnosed Atrial Fibrillation

**DOI:** 10.3390/biomedicines11010176

**Published:** 2023-01-10

**Authors:** Leon Blöbaum, Marco Witkowski, Max Wegner, Stella Lammel, Philipp-Alexander Schencke, Kai Jakobs, Marianna Puccini, Daniela Reißner, Daniel Steffens, Ulf Landmesser, Ursula Rauch, Julian Friebel

**Affiliations:** 1Department of Cardiology, Charité—Universitätsmedizin Berlin, Corporate Member of Freie Universität Berlin and Humboldt-Universität zu Berlin, 12203 Berlin, Germany; 2Department of Cardiovascular & Metabolic Sciences, Lerner Research Institute, Cleveland Clinic, Cleveland, OH 44195, USA; 3DZHK (German Centre for Cardiovascular Research), Partner Site Berlin, 10785 Berlin, Germany; 4Berlin Institute of Health at Charité—Universitätsmedizin Berlin, Charitéplatz 1, 10117 Berlin, Germany; 5Department of Cardiac Anesthesiology and Intensive Care Medicine, German Heart Center Charité, 13353 Berlin, Germany

**Keywords:** atrial fibrillation, leaky gut, microbial translocation, LPS, endotoxin, endotoxaemia, heart failure, fibrosis, gut-heart axis, biomarkers

## Abstract

Background: According to the leaky gut concept, microbial products (e.g., lipopolysaccharide, LPS) enter the circulation and mediate pro-inflammatory immunological responses. Higher plasma LPS levels have been reported in patients with various cardiovascular diseases, but not specifically during early atrial fibrillation (AF). Methods: We studied data and blood samples from patients presenting with first-diagnosed AF (FDAF) (*n* = 80) and 20 controls. Results: Circulating biomarkers that are suggestive of mucosal inflammation (zonulin, mucosal adhesion molecule MAdCAM-1) and intestinal epithelium damage (intestinal fatty acid binding protein, IFABP) were increased in the plasma of patients with FDAF when compared to patients with chronic cardiovascular diseases but without AF. Surrogate plasma markers of increased intestinal permeability (LPS, CD14, LPS-binding protein, gut-derived LPS-neutralising IgA antibodies, EndoCAbs) were detected during early AF. A reduced ratio of IgG/IgM EndoCAbs titres indicated chronic endotoxaemia. Collagen turnover biomarkers, which corresponded to the LPS values, suggested an association of gut-derived low-grade endotoxaemia with adverse structural remodelling. The LPS concentrations were higher in FDAF patients who experienced a major adverse cardiovascular event. Conclusions: Intestinal barrier dysfunction and microbial translocation accompany FDAF. Improving gut permeability and low-grade endotoxaemia might be a potential therapeutic approach to reducing the disease progression and cardiovascular complications in FDAF.

## 1. Introduction

Inflammatory immune processes are involved in the pathogenesis of atrial fibrillation (AF). AF occurs in the clinical setting of acute inflammation (critical illness) but more frequently in the chronic low-grade inflammation setting. An individual’s susceptibly to developing AF depends on the burden of risk factors that trigger chronic inflammation and mediate atrial myopathy (AM). AM is the substrate of AF that involves complex electrical and structural remodelling. Atrial fibrosis is an important structural correlate of AM. Chronic inflammation promotes collagen accumulation, which in turn affects conduction properties [1,2,3,4].

In recent years, the gastrointestinal tract has emerged as an important contributor to cardiovascular diseases. The concept of the gut-heart axis describes a link between gut pathology, the gut microbiome and cardiovascular disease. In this context, dysbiosis is characterised by a quantitative and or qualitative disruption of the microbiome. Consequently, changes in their composition and metabolic activities occur; however, intestinal permeability can also be affected. Following the leaky gut concept, an increased transepithelial passage of pathogens across the intestinal epithelium occurs through the transcellular or paracellular pathways [5]. This non-physiological passage of gastrointestinal microflora (or their derived products) through the intestinal epithelial barrier results in systemic immune activation with chronic inflammation, which affects vascular function [5,6,7].

Dysbiosis has previously been described for patients with AF. Traditional cardiovascular risk factors are associated with a shift in microbiota composition, the enrichment of Gram-negative bacteria and the increased production of lipopolysaccharide (LPS). LPS, which is derived from the outer membrane of Gram-negative bacteria, may enter the circulation through the gut mucosa and contribute to AF by worsening chronic inflammation [5,8,9,10,11]. Two recent studies obtained in preclinical models (faecal microbiota transplantation) have linked dysbiosis and circulating LPS with the development of AF via the atrial activation of nucleotide-binding and oligomerisation domain receptor protein (NLRP)-3 inflammasome in the atria [12,13]. Therefore, an increase in epithelial permeability with subsequent microbial translocation (MT) and immune stimulation could be a potential mediator in the pathophysiology of AF. In this regard, elevated plasma levels of LPS have been associated with AF, especially in elderly patients as well as those with recurrent AF and incident risk for major adverse cardiovascular events (MACEs) [13,14,15,16]. These studies were conducted in patients with paroxysmal, persistent, long-standing persistent and permanent AF, but not explicitly in patients with first-diagnosed AF. The first diagnosis of AF (FDAF) unveils a vulnerable high-risk group of patients in whom secondary preventive efforts to modify the course of the disease are critical [17].

Therefore, the present study aimed to assess biomarkers of intestinal barrier dysfunction and bacterial translocation and to correlate these with indicators of cardiac fibrosis and the onset of MACEs in a very early phase of AF.

## 2. Materials and Methods

### 2.1. Patient Studies

The local ethics committee (Charité—Universitätsmedizin Berlin) approved the study protocols, which were performed in accordance with the ethical principles in the Declaration of Helsinki. Each patient gave written informed consent before participating in the study. Routine laboratory results that are not mentioned separately in the methods section were obtained from medical records.

Our cohort consisted of 100 consecutive patients. This cohort has been already described previously [18]. Inclusion criteria for the AF group (*n* = 80) were as follows: age ≥ 18 years; willing to sign a written informed consent form; admission to our cardiology department due to a first documented diagnosis of AF. Exclusion criteria for the AF group were as follows: a reversible cause of AF (e.g., hyperthyroidism, acute myocardial infarction, myocarditis, pericarditis, acute infectious disease, acute inflammatory disease) or previous anti-coagulation. Baseline peripheral blood and data were collected directly within the first 24 h of hospitalisation. Data were collected from unselected patients who were available for sampling, with no adjustment of confounding [18].

During follow-up, a MACE was defined as the occurrence of cardiovascular death, unplanned re-hospitalisation for AF, unplanned hospitalisation for heart failure, a transient ischaemic attack (TIA), an ischaemic stroke, acute coronary syndrome (ACS), deep vein thrombosis or a peripheral thromboembolism.

The control group (*n* = 20) consisted of consecutive patients with a comparable cardiovascular risk profile (but without AF) who had been admitted to our cardiology department. These patients were hospitalised due to hypertensive heart disease, heart failure, or elective coronary angiography.

A detailed description of each patient’s characteristics is provided in Table 1. Furthermore, we considered gender-specific factors (Appendix A).

### 2.2. ELISA

Peripheral blood was collected in plasma tubes (Vacutainer, BD Biosciences, Heidelberg, Germany) and stored in polypropylene tubes at −80 °C until use. For zonulin, LPS (all purchased from Biomatik, Canada; EKC36091, EKC34448), LPS-binding-protein (LBP), soluble CD14 (sCD14), mucosal adhesion molecule MAdCAM-1, intestinal fatty acid binding protein (IFABP) (all purchased from R&D Systems, Minneapolis, MN, USA; DY870, DY383, DY3078, DY6056), endotoxin core antibodies (EndoCAb IgA, IgG, IgM) (Hycult Biotech, Netherlands; HK504-AGM), procollagen I C-terminal propeptide (PICP), procollagen I N-terminal propeptide (PINP), procollagen III N-terminal propeptide (PIIINP) and the C-telopeptide of type I collagen (ICTP) (all purchased from Aviva Systems Biology, San Diego, CA USA; OKEH00541, OKEH00679, OKEH00548, OKEH00680) ELISAs were performed according to the manufacturer’s instructions.

### 2.3. Statistical Analysis

Single comparisons were assessed using the Mann-Whitney U test. The Spearman coefficient was used for correlation analysis. All analyses were performed using GraphPad Prism version 9.3.0 software. Results are expressed as single values, mean ± SD. The overall α-level was 0.05.

## 3. Results

### 3.1. Biomarkers Suggest Intestinal Barrier Dysfunction in Patients with First-Diagnosed AF

To date, gut barrier function during early AF has not been studied. Abnormal intestinal permeability is attributed to mucosal inflammation and enterocyte apoptosis. Therefore, we analysed circulating biomarkers that are suggestive of mucosal inflammation and damage to the intestinal epithelium in a cohort of patients with first-diagnosed AF and control subjects. As such, elevated zonulin (which affects the integrity of the tight junctions between enterocytes) was detected in the plasma of patients with FDAF (Figure 1A) [19].

Mucosal inflammation is associated with the upregulation of chemotactic adhesion molecules like MAdCAM-1 [20,21]. In this study, we found that the plasma levels of gut-derived MAdCAM-1 were higher in patients with early AF when compared to the control group of patients with chronic cardiovascular diseases (CVDs) but without AF (Figure 1A).

Furthermore, increased circulating IFABP, which is released from damaged enterocytes, was present in the AF group when compared to controls (Figure 1B) [22].

### 3.2. Biomarkers Suggest Elevation of Circulating Gut-Derived Endotoxin

Following the leaky gut concept, microbial products enter the circulation and mediate a pro-inflammatory immunological response [5]. Elevated levels of LPS (and its ligands sCD14 and LBP), together with decreased levels of LPS-neutralising EndoCAbs in the serum/plasma, are well-defined surrogate markers of increased intestinal permeability and MT [6,23].

In our cohort of patients with FDAF, increased circulating markers of MT corroborated the biological significance of the presumed mucosal barrier defect (Figure 2).

As the prototype of a gut-derived, translocated microbial product, LPS was found to be increased in the plasma of our AF cohort (Figure 2A). In response to endotoxaemia, sCD14 and LBP were elevated (Figure 2B), whereas LPS-neutralising EndoCAbs were consumed and thus reduced in samples with enhanced endotoxin loads (Figure 2C,D) [6]. In patients with FDAF, the EndoCAb titres of mucosal-derived IgA were lower than those of the non-AF patients (Figure 2C). Furthermore, we detected a higher consumption (reduction) of EndoCAb IgG when compared to EndoCAb IgM, as suggested by a decreased IgG/IgM titres ratio in patients with early AF when compared to the controls, which suggests a rather chronic process (Figure 2D).

### 3.3. Increased Plasma Marker of Collagen Turnover in Patients with First-Diagnosed AF

Cardiac fibrosis is a pathogenic hallmark of AM. Structural remodelling due to an increase in collagen deposition is associated with the onset and progression of AF [1,2,3,4]. In patients that clinically present with a new onset of AF, biomarkers of collagen turnover suggest a pro-fibrotic state (Figure 3). N- and C-terminal pro-peptides are released during collagen synthesis and are indicative of cardiac dysfunction [24,25,26,27]. Accordingly, during early AF, we found higher plasma values of these pro-fibrotic markers (PICP, PINP, PIIINP) (Figure 3A). On the other hand, the circulating levels of collagen telopeptide (ICTP), which reflects collagen degradation [28], were lower in the AF cohort when compared to the controls (Figure 3B).

### 3.4. Circulating Endotoxin Associates with Biomarkers of Collagen Turnover in Patients with First-Diagnosed AF

Thus far, we have demonstrated that biomarkers suggest intestinal barrier dysfunction as well as increased MT and collagen turnover in patients with FDAF. Next, we evaluated the association of endotoxaemia with collagen metabolism to link chronic inflammation with structural remodelling.

The surrogate markers of collagen synthesis positively correlated with the plasma levels of LPS (Figure 4A). In contrast, ICTP, a biomarker of collagen degradation, was inversely associated with circulating LPS (Figure 4B). This suggests a possible link between disrupted intestinal homeostasis and AM.

### 3.5. Circulating Endotoxin Associates with the Onset of MACEs after First Diagnosis of AF

The MT product LPS has been linked to adverse outcome events in patients with CVDs, and higher levels of pro-inflammatory cytokines are known to induce a procoagulant state [15,16,29,30,31,32]. Therefore, we tested the hypothesis that endotoxaemia is specifically associated with the onset of MACE after FDAF.

In our cohort of patients with FDAF who experienced MACEs, the baseline plasma levels of LPS were higher when compared to patients without MACEs during follow-up (Figure 5A).

The disease progression of AF and AM (as expressed by unplanned re-hospitalisation for AF and unplanned hospitalisation for heart failure) corresponded to the plasma concentration of LPS (Figure 5B).

Patients that developed thromboembolic events (TIA and ischaemic stroke) and atherothrombotic complications (ACS) after FDAF had initially higher levels of LPS (Figure 5C).

The occurrence of cardiovascular death was also related to the LPS plasma levels (Figure 5D) during early AF.

## 4. Discussion

### 4.1. Intestinal Barrier Dysfunction in Patients with First-Diagnosed AF

Dysbiosis, impaired mucosal barrier function, and increased MT characterise the broad spectrum of CVDs [5]. Three main mechanisms are responsible for increased intestinal permeability: disrupted intestinal epithelial cell turnover due to increased apoptosis and/or insufficient regeneration, alterations of the tight-junction composition that enhance paracellular permeability for macromolecules, and the reduced clearance capacity of the mucosal immune system. These mechanisms can be initiated and expedited by chronic inflammatory stimuli [6,33]. Cardiovascular risk factors and established CVD were reported contributors to this process [5,9,34].

Previous studies have highlighted that mucosal inflammation in patients with primarily intestinal manifesting conditions (e.g., inflammatory bowel disease) increases the risk for a new onset of AF [35,36]. However, mucosal inflammation is not a prerequisite for developing AM or AF per se [37].

Nevertheless, our biomarker analysis suggested that, in the early phase of AF (FDAF), intestinal inflammation and enterocyte damage are present. To date, no direct evidence from functional studies for impaired mucosal integrity in patients with AF has been reported. However, there is conclusive evidence suggesting that dysbiosis is an important pathophysiological aspect of AF [9,10,11]. Dysbiosis affects MT directly (shift in microbial composition with an increase in antigen load) and indirectly (via mediating intestinal inflammation, which increases permeability) [5].

The most prominent indicator of gut dysbiosis and MT in AF is the presence of the circulating metaorganismal metabolite trimethylamine N-oxide (TMAO). Notably, its nutrient precursors are more abundant in a Western diet. Elevated plasma levels of TMAO have been reported in various stages of AF. TMAO was suggested to play an important role in the pathogenesis of AF, especially in patients with a high burden of cardiovascular risk factors and/or diseases. In preclinical models of AF, TMAO exacerbated autonomic activity and was responsible for the release of inflammatory and pro-fibrotic cytokines [5,9,10,11,38,39]. Notably, elevated plasma levels of TMAO or its precursors prospectively predicted the new onset of AF [40,41,42]. These data point towards a significant role for gut barrier dysfunction in the pathogenesis of AM and its progression to AF.

### 4.2. Increased MT in Patients with First-Diagnosed AF

Here, we highlight direct and indirect evidence of low-grade endotoxaemia during FDAF. Elevated LPS plasma levels have been previously reported in patients with AF, but not explicitly during FDAF. In these studies, the median LPS plasma levels ranged from 50 to 56 pg/mL [13,14,15,16]. The prevalence of AF was related to circulating LPS, with the highest prevalence of AF in patients with LPS values > 83 pg/mL [13]. In our cohort of FDAF patients, the median LPS was 85 pg/mL. Notably, low-grade endotoxaemia characterises the broad spectrum of CVDs. It is defined as circulating LPS levels of >20 pg/mL. Additionally, the definition of low-grade endotoxaemia requires the absence of sepsis criteria [29].

LPS is derived from the outer membrane of Gram-negative bacteria. It has been reported that dysbiosis in AF patients changes the microbiota composition towards potential LPS suppliers. As such, *Escherichia coli* was the most abundant pathogenic bacterial species found in AF patients [39]. Metagenomic data analysis also revealed the enrichment of LPS synthesis in the gut microbiota of AF patients [43]. The consumption of mucosal-derived EndoCAbs IgA, as observed in our study, suggests that a leaky gut significantly contributes to low-grade endotoxaemia in FDAF. Based on our reported reduction in EndoCAbs IgG/IgM ratio (compared to controls), we suggest that LPS exposure might be the result of rather chronic exposure to LPS. Likewise, an acute challenge with LPS in a human endotoxin model (LPS administration as an intravenous bolus of 2 ng/kg) did not result in new-onset AF [44].

### 4.3. Clinical Implications

#### 4.3.1. Low-Grade Endotoxaemia in Patients with First-Diagnosed AF Associates with Cardiac Fibrosis

An increase in epithelial permeability with subsequent MT and immune stimulation could be a potential mediator in the pathophysiology of AF.

Cardiac fibrosis is the underlying substrate of AM that subsequently predisposes patients to the occurrence of AF by altering conduction properties [1]. Chronic low-grade inflammation is a potential mediator of pro-fibrotic pathways [2]. After MT, LPS enters the circulation and mediates its pro-inflammatory properties via interaction with the innate immune toll-like receptor 4 (TLR4), LBP and sCD14 [29]. Our data suggest that endotoxaemia contributes to adverse cardiac structural remodelling.

Two recent preclinical models have causally linked dysbiosis, abnormal intestinal permeability and endotoxaemia to the pathogenesis of AF. It was demonstrated that LPS induced adverse atrial remodelling (fibrosis) and susceptibility to AF via activation of the TLR4/NF-κB/NOD-like receptor protein (NLRP)-3 inflammasome signalling pathway [12,13]. Activation of TLR4-signalling and associated inflammatory and fibrotic pathways has also been shown in patients with AF [45,46,47,48,49]. Low-voltage areas in AM are related to structural remodelling (cardiac fibrosis) [2]. It was demonstrated that monocytic TLR4 expression was positively correlated with low-voltage zones in AF patients [50]. Likewise, targeting the TLR4/NF-κB/NLRP3-axis reduced atrial fibrosis in preclinical models of AF [51,52]. Besides mediating inflammation-related fibrosis, LPS has been shown to induce atrial arrhythmogenesis via the downregulation of L-Type Ca^2+^ channels [53].

Our results suggest that in newly diagnosed AF, an LPS-induced pro-inflammatory vascular phenotype is associated with atrial adverse remodelling and contributes to the development of AF. Thus, novel strategies that target low-grade inflammation could be of interest in the prevention of AF. One such approach could be the widely used anti-diabetic drug metformin, which was shown to reduce pro-inflammatory cytokine production and thrombosis potential through the tissue factor pathway [32,54,55]. A recent study employing network medicine approaches and functional analyses in human inducible pluripotent stem cell (iPSC)-derived atrial cardiomyocytes suggested metformin as a repurposed drug candidate for AF [56].

#### 4.3.2. Low-Grade Endotoxaemia in Patients with First-Diagnosed AF Associates with MACEs during Follow-Up

AF is a chronic and progressive disease. The transition from subclinical to clinical AF (resulting in FDAF) depends on the burden of the electrical and structural atrial remodelling. Following FDAF, the occurrence of MACEs determines the individual prognosis [1]. Previous studies (not conducted explicitly in patients with early AF) reported that low-grade endotoxaemia (>50 pg/ml) was associated with an enhanced risk of MACEs [15,16]. Here, we demonstrated that patients with higher levels of LPS at FDAF were more likely to experience a MACE during follow-up, including a higher rate of cardiovascular death.

It has been reported that circulating LPS levels were associated with the recurrence of AF after radiofrequency ablation by increasing systemic inflammation and atrial fibrosis [14]. We suggest that chronic LPS-triggered low-grade inflammation contributed to AF progression following FDAF. The worsening of AM leads to the progression of AF from FDAF to advanced stages, with an increase in the burden of the AF duration and the density of episodes [1]. Furthermore, the presence of AF directly and indirectly impairs left ventricular function, leading to the development of HF [57,58]. Consequently, re-hospitalisations due to AF progression (attempts to restore sinus rhythm) and decompensated HF occurred, as observed in our study.

Acute atherothrombotic and thromboembolic adverse events are important contributors to AF-related morbidity and mortality [1]. It has been shown that atrial thrombogenesis (which causes thromboembolic stroke) and arterial thrombosis (causes ACS) were related to increased LPS-associated signalling via endothelial and platelet TLR4 [19,29,59]. Hence, we demonstrated that the occurrence of TIA, ischaemic stroke and ACS after FDAF were related to higher baseline levels of circulating LPS in our cohort.

### 4.4. Limitations

Although the first diagnosis of AF can be precisely defined, the individual AF duration is obscure. Therefore, based on our data, we cannot conclude at what point in time intestinal barrier dysfunction and MT are involved in the pathogenesis of AM and the onset of AF. Patients with AF were studied in a retrospective analysis with inherent biases and limitations [18]. Gut integrity can be assessed in vivo, by measuring the presence of molecular probes in the urine or blood after oral intake, and ex vivo in Ussing chambers combining electrophysiology and measurement of permeability using fluorescent probes of different molecular sizes. The latter is limited in the setting of AF due to the need for anti-coagulation. Furthermore, it will still not be possible to judge from where in the body the circulating LPS originate [60]. We used biomarkers as a valuable research tool to focus on one pathophysiological aspect (low-grade endotoxemia) of the gut-heart axis. Prospective trials that account for the complexity of the gut-heart axis and include different techniques (e.g., stool samples, biopsies, metabolomics, biomarkers) would be necessary.

## 5. Conclusions

Intestinal barrier dysfunction and MT accompany FDAF. The detection of low-grade endotoxaemia identifies patients that are at a high risk of adverse events. This may help to improve the risk stratification of patients with FDAF. Dietary interventions can rapidly shift the microbial composition in the gut; however, the clinical relevance of these therapeutic options needs to be determined. Improving gut permeability and low-grade endotoxaemia might be a potential therapeutic approach to reducing disease progression and cardiovascular complications in FDAF.

## Figures and Tables

**Figure 1 biomedicines-11-00176-f001:**
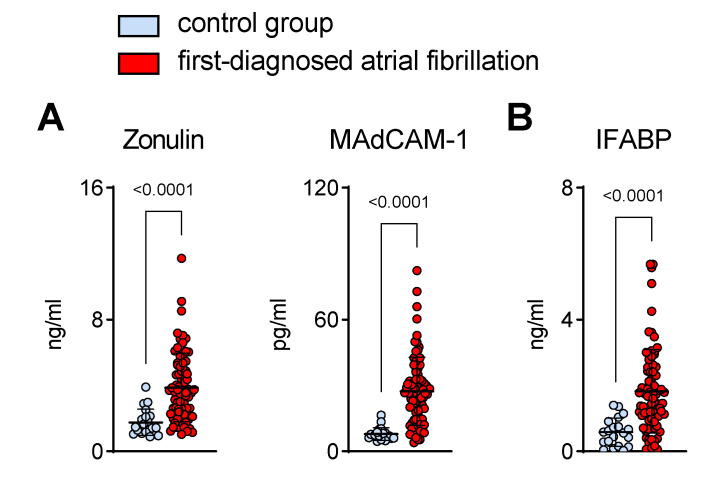
Intestinal barrier dysfunction in patients with first-diagnosed AF. Elevated circulating biomarkers suggestive of intestinal inflammation (**A**) (zonulin and mucosal endothelial cell adhesion molecule, MAdCAM-1) and enterocyte damage (**B**) (intestinal fatty acid binding protein, IFABP) that are associated with abnormal intestinal permeability. Patients with first-diagnosed AF (*n* = 80) were compared to controls (patients with chronic cardiovascular diseases but without AF) (*n* = 20). Results are expressed as single values, mean ± SD, and *p*-values.

**Figure 2 biomedicines-11-00176-f002:**
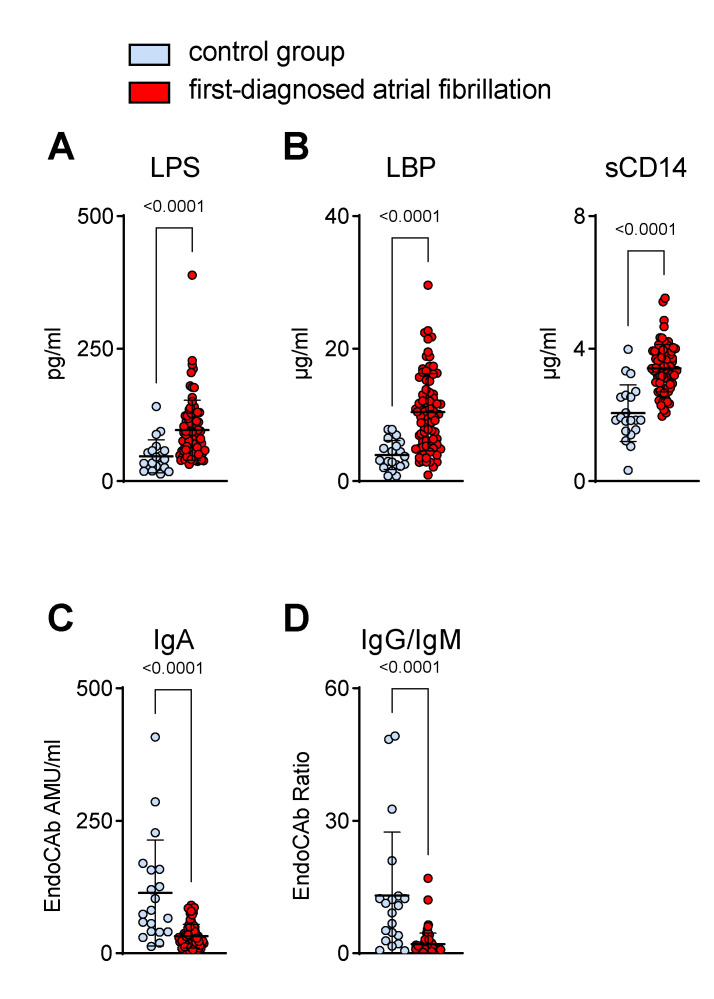
Biomarkers suggest the elevation of circulating gut-derived endotoxin in patients with first-diagnosed AF. Concentrations of markers for increased microbial translocation in plasma were measured by ELISA. Elevated levels of direct (lipopolysaccharide, LPS) (**A**) and indirect (LPS-binding protein, LBP; soluble CD14, sCD14) (**B**) indicators of low-grade endotoxaemia were present in plasma during early AF. Lower titres of circulating endotoxin core IgA antibodies (EndoCAb) (**C**) and a reduced ratio of EndoCAb IgG/IgM titres (**D**) in the AF cohort compared to controls. Results are expressed as single values (*n* = 20/80), mean ± SD, and *p*-values.

**Figure 3 biomedicines-11-00176-f003:**
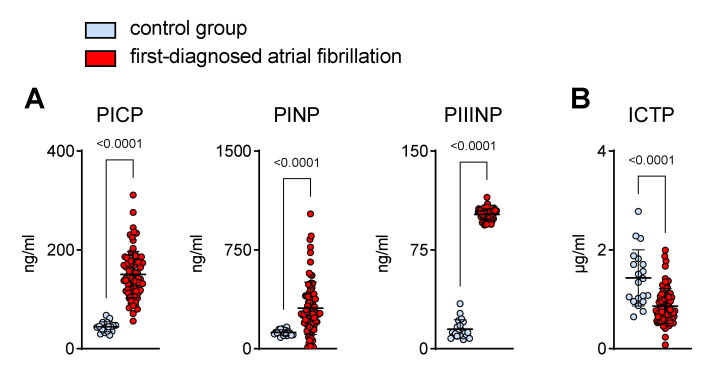
Increased plasma markers of collagen turnover in patients with first-diagnosed AF. Elevated surrogate markers of collagen synthesis (procollagen I C-terminal propeptide, PICP; procollagen I N-terminal propeptide, PINP; procollagen III N-terminal propeptide, PIIINP) (**A**) and reduced collagen degradation fragments (C-telopeptide of type I collagen, ICTP) (**B**) in the plasma of early AF patients (*n* = 80) vs. controls (*n* = 20). Results are expressed as single values, mean ± SD, and *p*-values.

**Figure 4 biomedicines-11-00176-f004:**
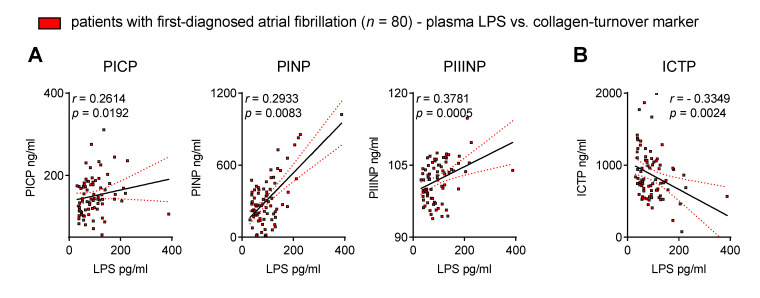
Circulating endotoxin associates with biomarkers of collagen turnover in patients with first-diagnosed AF. Elevated lipopolysaccharide (LPS) correlates with surrogate markers of collagen synthesis (procollagen I C-terminal propeptide, PICP; procollagen I N-terminal propeptide, PINP; procollagen III N-terminal propeptide, PIIINP) (**A**) and collagen degradation (C-telopeptide of type I collagen, ICTP) (**B**) in the plasma of early AF patients vs. controls. Results are expressed as single values (*n* = 80), Spearman correlation coefficients and linear regression lines (black full lines) with 95% CI (red dotted lines).

**Figure 5 biomedicines-11-00176-f005:**
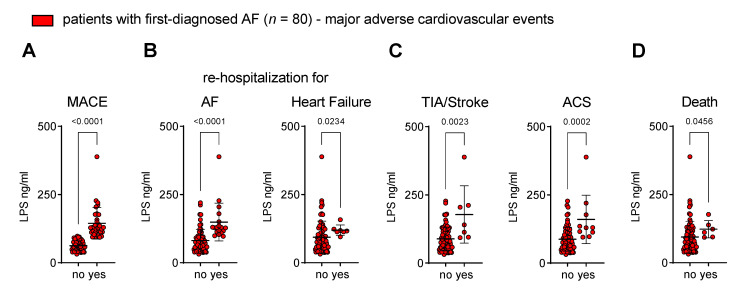
Circulating endotoxin associates with the onset of major adverse cardiovascular events after first diagnosis of AF. Patients were stratified into two groups according to the occurrence of MACE (no vs. yes) during follow-up after their first diagnosis of AF. (**A**) Composite endpoint MACE: first occurrence of cardiovascular death, unplanned re-hospitalisation for AF, unplanned hospitalisation for heart failure (HF), transient ischaemic attack (TIA), ischaemic stroke, acute coronary syndrome (ACS), deep vein thrombosis, peripheral thromboembolism. Adverse outcome events that are associated with disease progression of AF (unplanned re-hospitalisation for AF, unplanned hospitalisation for HF) (**B**), thromboembolic and atherothrombotic complications (TIA, ischaemic stroke, ACS) (**C**) and cardiovascular death (**D**) are related to circulating LPS at time of first diagnosis of AF. Results are expressed as single values (*n* = 80), mean ± SD, and *p*-values.

**Table 1 biomedicines-11-00176-t001:** Baseline characteristics of patients with first-diagnosed AF and controls.

	Control Group	Patients with First-Diagnosed AF	
	(*n* = 20)	(*n* = 80)	*p*-Value
Male/Female	50%/50%	62.5%/37.5%	n.s.
CHA_2_DS_2_-VASc	3.45	3.98	n.s.
History of Heart Failure	20%	26%	n.s.
Hypertension	85%	87.5%	n.s.
Age (years)			
<65	45%	27.5%	n.s.
65–75	30%	32.5%	n.s.
>75	25%	40%	n.s.
Diabetes	25%	30%	n.s.
History of TIA/Stroke	5%	10%	n.s.
Body Weight (kg)	82.95	85.93	n.s.
BMI kg/m^2^	27.97	27.55	n.s.
Echocardiography			
Ejection Fraction	62%	59%	n.s.
LVEDD (mm)	52	49	n.s.
E/e‘_sept_	10.24	10.59	n.s.
E/e‘_lat_	6.86	9.11	n.s.
IVSd (mm)	11	10.76	n.s.
LVPWd (mm)	10.54	10.38	n.s.
LA volume (ml)	65.88	61.88	n.s.
TAPSE (cm)	2.4	2.35	n.s.
sPAP	32.75	33.05	n.s.
FDAF			
Spontaneous conversion	-	33%	
Electrical cardioversion	-	67%	
Hospital discharge in SR	-	100%	
Follow-up			
years	1	2.99	
MACE	0%	41.25%	

Values are mean or %. Abbreviations: TIA, transient ischaemic attack; LVEDD, left ventricular end-diastolic diameter; E/e′_sept_, ratio between early mitral inflow velocity and septal early diastolic tissue velocity; E/e′_lat_, ratio between early mitral inflow velocity and lateral early diastolic tissue velocity; IVSd, interventricular septum diameter; LVPWd, left ventricular posterior wall thickness in diastole; LA volume, left atrial volume; sPAP, systolic pulmonary artery pressure; SR, sinus rhythm; MACE, major adverse cardiovascular events; n.s., not significant.

## Data Availability

Data from patients are not publicly available due to general data protection regulations.

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
