# Peer review of "Intestinal Barrier Dysfunction and Microbial Translocation in Patients with First-Diagnosed Atrial Fibrillation"

_biomedicines, 2023, doi:10.3390/biomedicines11010176_

Round 1

Reviewer 1 Report

his article addresses an extremely interesting topic that has not yet been properly recognized by the medical community. The authors used a variety of valuable biomarkers in the study in order to confirm the hypothesis. I think the study is well-designed and has an up-to-date bibliography.
There are some questions that should be made about the manuscript:
1) Сould you please provide the information about the age and body mass index of patients with AF and controls
2) During the follow-up, how many patients with first-diagnosed AF had developed persistent, long-standing persistent and permanent AF?
3) Could you please provide some echocardiographic parameters of patients with AF and controls if it possible.

Author Response

Reviewer #1:

Comment 1:

Сould you please provide the information about the age and body mass index of patients with AF and controls.

Reply

Thank you for catching this. We now have added this information to the baseline characteristics.

Comment 2:

During the follow-up, how many patients with first-diagnosed AF had developed persistent, long-standing persistent and permanent AF?

Reply:

The reviewer raises a very interesting question. 1/3 of patients converted into SR spontaneously. All FDAF patients were discharged in SR after their first hospitalization. We have now added this information to the baseline characteristics.

Inflammation and atrial fibrosis are fundamental mechanisms of the pathophysiological processes involved in the development of AM and AF. The degree of atrial inflammation and fibrosis correlates with the susceptibility to AF, increases the frequency of AF paroxysms and increases the likelihood of progression to persistent and permanent AF. In our study we demonstrated that during the follow-up of ≈ 3 years, 18 FDAF patients were re-hospitalized due to AF (electrical cardioversion). This subgroup was characterized by higher plasma levels of LPS compared to FDAF that were not re-hospitalized during the follow-up. This suggests that low grade endotoxemia might influence the course of AF. On the other hand, different rhythm control strategies, that also include ablation procedures, might have a strong impact on this outcome event. Therefore, we authors decided not to subdivide into the different AF types. This should be considered in future trials. We have discussed this in the new limitations section of the manuscript.

Comment 3:

Could you please provide some echocardiographic parameters of patients with AF and controls if it possible.

Reply:

We agree. We now have added this information to the baseline characteristics.

Reviewer 2 Report

Present study by Blöbaum et al., aimed at deciphering the possible role of intestinal barrier dysfunction and microbial translocation in first diagnosed atrial fibrillation (FDAF). They basically used some already known biomarkers of gut intestinal permeability dysfunction to find if these markers are altered in FDAF compared to non-AF control individuals. However, this reviewer finds major concerns detailed below due to which it can not be accepted in its current form.

1. The basic issue with this study is uneven inclusion of the controls compared to the FDAF patients. On one hand, there are only 20 controls against 80 patients, on the other hand, the gender distribution is completely unmatched which raises serious doubts on the presented findings. In control group (N=20), 50% are males and 50% are females, whereas, in FDAF group, there are 62.5% males (i.e., 50 patients), and 37.5% females (i.e., 30 patients). Ideally, control subjects should have been recruited matching with the patient group for sex, age, BMI, hypertension, and diabetes history to have unbiased outcome. But with such a big disparity between the number of controls Vs patients, one of the most important factor for clinical studies is not accounted for.

2. Authors extrapolated their results claiming microbial translocation in FDAF patients only on the basis of plasma parameters without providing any other supporting data, which this author believes is very preliminary.

Author Response

Reviewer #2:

Comment 1:

The basic issue with this study is uneven inclusion of the controls compared to the FDAF patients. On one hand, there are only 20 controls against 80 patients, on the other hand, the gender distribution is completely unmatched which raises serious doubts on the presented findings. In control group (N=20), 50% are males and 50% are females, whereas, in FDAF group, there are 62.5% males (i.e., 50 patients), and 37.5% females (i.e., 30 patients). Ideally, control subjects should have been recruited matching with the patient group for sex, age, BMI, hypertension, and diabetes history to have unbiased outcome. But with such a big disparity between the number of controls Vs patients, one of the most important factor for clinical studies is not accounted for.

Reply:

We thank the reviewer in supporting our intention and we understand the reviewer’s concern about the potential risk of bias. We already stated in the methods that we included 100 consecutive patients. During the recruitment phase (in which we recruited 80 patients with FDAF), we only could consider twenty controls. All other patients were admitted due to ACS, decompensated heart failure etc. In order with your suggestion, we now stated in the methods section that data were collected data from unselected patients who were available for sampling, with no adjustment of confounding. We refrained from including healthy controls because the authors wanted to study effects beyond the traditional risk factors for AF.

We completely agree with the reviewer that small sample sizes always raise concerns about statistical power. However, this basic assumption would be challenged by ethical restrictions. The data obtained in our study, now will provide sufficient evidence to conduct larger trials.

As suggested by reviewer #2, we have restructured and extended the baseline characteristics. We did not find statistically significant differences especially for the distribution of age (according to the CHA2DS2-VASc-Score), BMI, hypertension, and diabetes history. Therefore, in the opinion of the authors, our cohort represents a typical risk constellation for the development of AF.

The reviewer raises also a very interesting question. Gender aspects are very important in the pathogenesis of AM and AF. In order with your suggestion, we now have added new supplementary figures (S1-S3) to account for gender aspects.

Nevertheless, we cannot exclude the risk of bias or confounding. Therefore, we now have discussed this limitation to address your concerns (new separate limitations section).

Comment 2:

Authors extrapolated their results claiming microbial translocation in FDAF patients only on the basis of plasma parameters without providing any other supporting data, which this author believes is very preliminary.

Reply:

Several statements that we made were more ambiguous than intended, and we have adjusted the text to be clearer. Our study was mainly based on circulating biomarkers of intestinal barrier dysfunction and microbial translocation. However, there is no gold standard to measure intestinal barrier function. Gut integrity can be assessed in vivo, by measuring the presence of molecular probes s in urine or blood after oral intake. Whilst this method can determine overall differences in permeability between the small and large intestine, it is unable to reveal detailed information relating to precise regional locations of altered integrity within each organ and also lacks standardization. Additionally, external factors including gastrointestinal motility and mucosal blood flow may lead to inaccuracy of permeability measurements. The Ussing system (mounting biopsies in Ussing chambers) offers an ex vivo measurement of permeability using fluorescent probes as well as electrophysiological measurements (although the risk of bleeding must be considered). Furthermore, only a minority of the different anatomical sites can be easily reached for routine endoscopy. Due to these limitations, the authors decided to focus on approved circulating biomarker.

The authors agree that it is important to combine different techniques in order to give an accurate picture of the intestinal barrier as possible. We agree that additional analyses would provide useful and important data, but we believe that the recommended analyses are outside the scope of this study.

However, our intention was giving a first description of the overall mucosal integrity and low-grade endotoxemia in this specific clinical situation of FDAF, thereby providing a rational for the conduction of larger (prospective) trials. In vivo, ex vivo or biomarkers in the specific context of FDAF have not been studied before. In the opinion of the authors, the first diagnosis of AF (FDAF) unveils a vulnerable high-risk group of patients in whom secondary preventive efforts to modify the course of the disease are critical. In order with your suggestions, we now have discussed this important aspect in the new limitations section.

Round 2

Reviewer 1 Report

I approve revised vesrion of paper.

Author Response

We thank the reviewer for time and effort assessing the previous version of the manuscript.

Reviewer 2 Report

Please provide experimental details, maufacturer details for the ELISA or other kits used for biomarker measurements. Just mention of ‘experiments are performed as per the manufacturerr’s instructions’ is not sufficient.

Author Response

We thank the reviewer for time and effort assessing the previous version of the manuscript.

Comment:

Please provide experimental details, maufacturer details for the ELISA or other kits used for biomarker measurements. Just mention of ‘experiments are performed as per the manufacturerr’s instructions’ is not sufficient.

Reply:

Thank you for catching this glaring and confusing error, which we have now corrected in the manuscript. All ELISA were performed according to the standardized protocols of the manufacturers which are which are freely accessible.